# The Profiles and Functions of RNA Editing Sites Associated with High-Altitude Adaptation in Goats

**DOI:** 10.3390/ijms24043115

**Published:** 2023-02-04

**Authors:** Li Li, Xiaoli Xu, Miao Xiao, Chunhua Huang, Jiaxue Cao, Siyuan Zhan, Jiazhong Guo, Tao Zhong, Linjie Wang, Liu Yang, Hongping Zhang

**Affiliations:** Farm Animal Genetic Resources Exploration Innovation Key Laboratory of Sichuan Province, College of Animal Science and Technology, Sichuan Agricultural University, Chengdu 611130, China

**Keywords:** goat, RNA editing, functional characterization, high-altitude adaptation

## Abstract

High-altitude environments dramatically influenced the genetic evolution of vertebrates. However, little is known about the role of RNA editing on high-altitude adaptation in non-model species. Here, we profiled the RNA editing sites (RESs) of heart, lung, kidney, and longissimus dorsi muscle from Tibetan cashmere goats (TBG, 4500 m) and Inner Mongolia cashmere goats (IMG, 1200 m) to reveal RNA editing-related functions of high-altitude adaptation in goats. We identified 84,132 high-quality RESs that were unevenly distributed across the autosomes in TBG and IMG, and more than half of the 10,842 non-redundant editing sites were clustered. The majority (62.61%) were adenosine-to-inosine (A-to-I) sites, followed by cytidine-to-uridine (C-to-U) sites (19.26%), and 32.5% of them had a significant correlation with the expression of catalytic genes. Moreover, A-to-I and C-to-U RNA editing sites had different flanking sequences, amino acid mutations, and alternative splicing activity. TBG had higher editing levels of A-to-I and C-to-U than IMG in the kidney, whereas a lower level was found in the longissimus dorsi muscle. Furthermore, we identified 29 IMG and 41 TBG population-specific editing sites (pSESs) and 53 population-differential editing sites (pDESs) that were functionally involved in altering RNA splicing or recoding protein products. It is worth noting that 73.3% population-differential, 73.2% TBG-specific, and 80% IMG-specific A-to-I sites were nonsynonymous sites. Moreover, the pSESs and pDESs editing-related genes play critical functions in energy metabolisms such as ATP binding molecular function, translation, and adaptive immune response, which may be linked to goat high-altitude adaptation. Our results provide valuable information for understanding the adaptive evolution of goats and studying plateau-related diseases.

## 1. Introduction

The functions and mechanisms of RNA editing have been well studied, the functions of which include modifying the sequence of RNA precursors [1,2], changing mRNA isoforms, recoding proteins, and the resulting physiological and pathological changes in organisms [3,4,5]. RNA editing frequently occurs in the noncoding regions and is significantly enriched in the 3′ untranslated regions (3′-UTR) [6], which play a central role in RNA degradation and nuclear retention [7,8,9]. Adenosine-to-inosine (A-to-I) RNA editing is mediated by adenosine deaminase acting on the RNA (ADAR) family, and cytidine-to-uridine (C-to-U) RNA editing is mediated by apolipoprotein B mRNA editing enzyme catalytic polypeptide (APOBECs) family. A-to-I and C-to-U editing are considered frequent and crucial regulatory mechanisms for the development of species [10,11]. For example, A-to-I editing of the *GluR2* transcript revealed that the substrate of ADAR2 is crucial for the normal development of the nervous system of zebrafish, ADAR1-deficient mice died around 12 days after birth, and ADAR1 deficiency caused A-to-I editing depletion at the A and B sites of 5-HT_2C_ receptor transcripts. [12,13].

RNA editing is affected by many factors, including temperature, tissue context, genotype, feeding conditions, and age [14,15], while RNA editing controls phenotype via editing the sequence of RNA molecules [16,17]. For example, most RNA editing sites (RESs), mainly mediated by the ADAR family, were tissue-specific and enriched in tissue-specific biological functions. In addition, the number of RESs showed a dynamic trend during skeletal muscle development. The correlation between editing levels of RESs and their gene expression has been identified [18,19]. Our previous study found a significant difference in RESs counts and editing levels at goat fetal and postnatal periods, whereas *SMAD3*, *SIAH1*, and *CDK13* had embedded functional editing sites and were involved in muscle development [17]. Furthermore, several studies have identified RESs in farm animals such as pigs [19,20,21,22], chickens [14], sheep [11], and goats [17]. For instance, the brain, subcutaneous fat, heart, liver, muscle, lung, and ovary of pigs had 74,863 RESs [22]. In addition, nine bovine tissues have identified 1600 novel A-to-I editing sites [18], which may be related to the tissue functions.

The species’ adaptation to local conditions has been promoted through generational inheritance [23]. In particular, the intense selective pressures (e.g., low oxygen, low temperature, and ultraviolet radiation) that alter the morphology [24,25], physiology [26], and genetics of animals that live on the plateau [27] have been well-studied. RNA editing, an essential resource for gene regulation, has been affected by environmental factors over generations [28]. The octopus could live in cold water temperatures owing to the A-to-I editing in the gene associated with ion channels [15]. However, little is known about RNA editing functions in high-altitude animal adaptation. The kidney, heart, lung, and longissimus dorsi muscle tissues are critical for high-altitude transformation. Here, we systematically compared RNA editing in these tissues in indigenous Tibetan cashmere goats (TBG) and Inner Mongolia Cashmere goats (IMG), which have a close genetic distance and share a common ancestor. These findings might shed new light on how mammals have evolved to live at high altitudes.

## 2. Results

### 2.1. The Landscape of RNA Editome in Goats

We identified an average of 5747 candidate RESs for 40 samples. After filtering the raw RNA editing sites, 84,132 high-quality RESs were identified (Appendix A), and 10,842 non-redundant RESs were detected across the goat genome (Figure 1A). Interestingly, these editing sites are unevenly distributed across the autosomes, with the highest editing frequency found in chr3 and chr19 and the lowest in chr26 and chr27 (Figure 1B). In addition, RESs were slightly raised at two ends on each chromosome (Figure 1C), and 6162 (56.83%) editing sites represented the nearest distance of less than 20 bp for each site pair (Figure 1D), which confirmed that RESs are clustered [29].

More than half (5920) of the non-redundant sites were detected in at least five samples, with 35 sites (0.32%) appearing in 40 samples (Figure 2A). The editing levels were enhanced to less than 0.5 (Figure 2B). In total, 36,295 (43.14%) of RESs had editing levels exceeding 0.5, and 20,251 (24.07%) sites had editing levels greater than 0.95, whereas 16,389 (19.48%) RESs were completely edited (Appendix A). Moreover, 3871 (35.70%) and 2883 (26.59%) of the non-redundant sites were detected in the DNA positive strand and reverse strand, while 5562 (51.30%) were ambiguous (Figure 2C). A-to-I (62.61%) was the most common type of RNA editing, followed by C-to-U (19.26%), according to the median counts of AG (36.13%), TC (26.48%) (regarded as AG in the opposite strand), CT (8.99%), GA (10.27%) (regarded as CT in the opposite strand) (Figure 2D and Appendix A). AG and TC editing levels were 0.31 and 0.36, respectively (Figure 2E). We further analyzed the A-to-I and C-to-U editing sites and obtained 11,705 A-to-I and 6436 C-to-U editing sites, located in 1389 and 838 non-redundant loci, which were from 811 protein-coding genes (PCGs), 16 lincRNAs, 14 pseudogenes, and 10 processed pseudogenes in the goat genome (Figure 2F and Appendix A). In PCGs, 1386, 810, 683, 115, and 24 A-to-I or C-to-U editing sites overlapped 2163 transcripts, 1170 exons, 966 coding regions (CDSs), 185 3′-UTRs, and 21 5′-UTRs, respectively (Figure 2G). Gene Ontology (GO) enrichment analysis showed that those PCGs mainly involved in microfilament motor activity, contractile fiber, myosin complex, myofibril, and sarcomere (Figure 2H and Appendix A).

### 2.2. Different Flanking Sequences, Amino Acid Mutations, and Alternative Splicing Activities Are Observed at A-to-I and C-to-U Sites in Goats

We first calculated the correlation coefficient between the editing levels of RESs (A-to-I and C-to-U sites) and the expression levels of catalyzing genes such as *ADAR*, *ARARB1*, *ADARB2*, *APOBEC1*, *APOBEC2*, and *APOBEC4* in goats. A total of 724 (32.5%) RESs significantly correlated with the expression of catalyzing genes, and 90.5% of them positively correlated, whereas 38.4% of C-to-U sites negatively correlated with the expression of *APOBEC2* (Figure 3A). Among these, 392 (28.2%), 309 (22.2%), and 40 (2.9%) A-to-I sites were correlated to *ADAR*, *ADARB1*, and *ADARB2*, respectively. In addition, 94, 104, and 127 C-to-U sites were related to the expression of *APOBEC1*, *APOBEC2*, and *APOBEC4*, respectively (Appendix A). This finding indicates that multiple characterizations might exist in a gene family member of RNA editing, which might be mediated by other regulators. We also performed a correlation analysis between RESs and their editing-related genes. After filtering out the low-expression genes (68), we discovered that the expressions of 317 PCGs, including *SFTPB*, *ATP2A1*, *EIF2AK4*, *KLF13*, *LDB3*, *SLC13A1*, and *ACMSD*, and the editing levels of 874 RESs (40.5%) were strongly correlated (Appendix A). The correlation coefficient ranged from 0.3 (including 21 negative correlations) to 0.8, accounting for 89.7% (565/630) A-to-I editing sites and 94.5% (241/255) C-to-U editing sites (Figure 3B).

To determine whether RNA editing sites are enriched in specific sequence motifs, we explored the base preference of RESs (1389 and 838 non-redundant A-to-I and C-to-U editing sites) and found a differing probability for their 10-bp flanking region (total of 21 bp) (Figure 3C). For the A-to-I editing sites, G was absent from one base upstream but enriched in one downstream. In addition, the Gs and Cs frequently appear beside A-to-I in no discernible pattern, whereas the C-to-U sites were more likely to be occupied by Gs and As. There is no base preference in each position of codons for 977 RESs in 748 CDSs encoding 686 proteins located in A-to-I or C-to-U editing sites (Figure 3D). The number of RESs (671) located at the third position of the codons was higher than at the first (404) and second positions (401), indicating that RNA editing generally causes a benign outcome since the third base of codons has a low probability of altering their coded amino acid (Appendix A). However, we discovered that the encoded amino acids changed vastly after editing, and A-to-I or C-to-U resulted in the opposite pattern of amino acids. A large part of serine, methionine, isoleucine, and lysine encoded from reference codons was edited to glycine and arginine by A-to-I editing. In contrast, C-to-U editing mainly turns arginine and glycine into methionine and serine (Figure 3E). These suggest that RNA editing changes the physical and chemical properties of the encoded protein to some extent.

To investigate RNA editing involved in alternative splicing of gene transcripts, 1228 RESs were evaluated in 867 exons of 849 transcripts from 552 genes (Appendix A). The distribution peak of the minimum distances from RESs to the exon boundary was close to 0, indicating that a large number of RESs occurring at the start or end of an exon may change the binding site of the spliceosome and alter the transcript isoform (Figure 3F). Among them, the minimal distances to the exon boundary of C-to-U editing (average of 76) were significantly lower than A-to-I editing sites (average of 105), implying that C-to-U editing may have a higher activity on alternative splicing than A-to-I editing in goat (Figure 3G).

### 2.3. TBG Shows Higher A-to-I and C-to-U Editing Levels than IMG in the Kidney

To systematically investigate the RNA editing profiling of four tissues (heart, lung, kidney, and longissimus dorsi muscle) from TBG and IMG, we generated a pairwise comparison across tissues and populations based on 1389 A-to-I and 838 C-to-U non-redundant editing sites (Figure 4A and Appendix A). The A-to-I and C-to-U sites were unevenly distributed across the tissues, with the majority found in the lung and kidney and fewer in the heart and longissimus dorsi muscle. We also found that almost every comparison for A-to-I and C-to-U sites showed a significant difference in editing counts and levels among tissues (Figure 4B,C), which shows the tissue-specific RNA editing pattern in goat populations. In addition, we compared the editing counts and levels between TBG and IMG for each tissue, and no significant difference was found in the counts of each tissue (Figure 4D). However, the editing levels of A-to-I and C-to-U in the kidney of TBG were significantly higher than IMG. The A-to-I editing level in the longissimus dorsi muscle between TBG and IMG also differed considerably (Figure 4E). These findings suggest that the RNA editing pattern in tissues is relatively more prominent than in populations, and there are higher editing levels present in the kidney of TBG than that of IMG, which potentially assists TBG in adapting to the environment.

### 2.4. The Kidney and Longissimus Dorsi Muscle Tissues Exhibit More Nonsynonymous A-to-I and C-to-U Sites in Goats

RESs specifically appearing in organs involve a tissue-specific process. We investigated the function of the tissue-specific RNA editing sites (tSESs). As shown in Figure 5A, a total of 681, 553, 459, and 534 RESs were detected in all four, three, two, and only one tissue, composed of 376, 297, 284, 432 A-to-I sites and 305, 256, 175, 102 C-to-U sites, respectively. Moreover, we identified tSESs in the heart (18), kidney (272), lung (68), and longissimus dorsi muscle (176) and selected 8, 148, 37, and 36 sites as tSESs for downstream analysis (Appendix A). The editing levels of 209 (178 A-to-I and 31 C-to-U sites) tSESs positively correlated with their gene expression, 29.2% A-to-I and 16.1% C-to-U tSESs presented a strong correlation with a coefficient over 0.8 (Figure 5B and Appendix A). Notably, the longissimus dorsi muscle exhibited the most significant proportion (47.6%) of high correlation between the editing level of A-to-I tSESs and its related gene expression (Figure 5C).

Based on 3, 19, 8, and 17 genes embedding 6, 139, 35, and 29 tSESs in the heart, kidney, lung, and longissimus dorsi muscle, respectively. We performed the GO enrichment analysis for each tissue-specific RNA editing-related gene. In the kidney, three tSESs-related genes (*SLC5A12*, *SLC13A1*, and *SLC22A4*) were significantly enriched in symporter activity and ion transport, which play a crucial role in maintaining homeostasis. In comparison, five muscle-related genes (*MYH8*, *MYH2*, *MYH3*, *NEB*, and *OBSCN*) function in contractile fiber and sarcomere (Figure 5D,E and Appendix A). Moreover, RNA editing events at nonsynonymous sites showed strong adaptive signals. Thus, we identified protein-coding changes caused by 30 A-to-I tSESs and 27 C-to-U tSESs in the CDSs of 18 proteins, respectively. We realized that over 80% A-to-I and 70% C-to-U tSESs could recode proteins (Figure 5F and Appendix A). In the kidney, we observed 85.7% A-to-I and 71.4% C-to-U nonsynonymous sites can recode proteins, whereas 84.2% A-to-I and 91.7% C-to-U in the longissimus dorsi muscle (Figure 5G). This demonstrates the tissue-specific pattern of RNA editing and adaptation signals in goat populations.

We then analyzed the tSESs involved in gene transcript splicing and found six tSESs less than 5 bp from the start or end of six exons in four genes (Appendix A). Interestingly, four tSESs (3 A-to-I and 1 C-to-U) in muscle and C-to-U tSESs in kidney were significantly associated with their editing-related genes expression, potentially recoding eleven amino acids, varying eight protein products, and altering seven transcript isoforms (Appendix A). An example of the A-to-I tSESs 19:29140761(-) located in the 16th exon of transcript MYH8-203 is shown in Figure 5H.

### 2.5. Population-Specific A-to-I and C-to-U RNA Editing Sites in TBG and IMG

We examined the population-specific editing sites (pSESs) and population-differential editing sites (pDESs) to define the function of RESs in population adaptation. The majority (78.1%) of A-to-I and C-to-U sites were shared between TBG and IMG, whereas 10.6% and 11.4% in IMG and TBG, respectively (Figure 6A). Then, we selected the sites supported by at least three samples and identified 29 IMG pSESs, 41 TBG pSESs, and 53 pDESs (Appendix A and Figure 6B), which overlapped 80 CDSs in 86 exons of 120 transcripts from 98 genes (*ZNF644*, *ATF6*, *ADAM10*, *OBSCN*, *MYCT1*, *OGFOD1*, *MTO1*, *ARAP2*, *SCP2*, *KIF17,* etc.) (Appendix A). Of them, 19 sites were correlated with the expression of 16 genes (Appendix A and Figure 6C), and 69 sites might recode 71 amino acids and change 46 protein products (Appendix A and Figure 6D). Additionally, 73.3% population-differential, 73.2% TBG-specific, and 80% IMG-specific A-to-I sites were nonsynonymous, which suggested IMG and TBG also have their unique regulatory pathways. We also analyzed the pSESs or pDESs located at 10 bp distance to the start or end of exons and found five sites may involve eight transcripts alternative splicing (Appendix A and Figure 6E). The editing levels of four pDESs, seven IMG pSESs, and eight TBG pSESs were significantly correlated with the expressions of 4, 6, and 6 genes separately (Figure 6F–H). Only one of these 16 editing-related genes was officially named *OBSCN*, a gene related to ATP binding molecular function, protein phosphorylation, and biological process (Figure 6I). We further investigated the functions of those 15 Ensembl genes by utilizing UniProt [30]. They were related to biological processes, including protein dephosphorylation, translation, antigen processing and presentation, immune response, and regulation of transcription (Appendix A). These results suggest that RNA editing involves high-altitude goat adaptation and is assigned to multiple biological processes.

## 3. Discussion

RNA editing plays a critical role in all aspects of RNA transcripts, including splicing, mRNA stability, and translation. A-to-I RNA editing, which is the most prominent RNA editing type [19,22], has been associated with human disorders [31,32,33,34] and is also essential for the growth of multiple species, including mice [35], chickens [36], bovines [18], and sheep [11]. Microsatellite [37,38], exome sequencing [39], and mitochondrial DNA sequence analysis [40] results showed that the genetic distance between TBG and IMG was close and had the same maternity [41]. Moreover, IMG was the closest to TBG in body weight and fleece performance (fleece yield, thickness, diameter). Climate and geographical factors had a great impact on goat breeds. Therefore, we characterized RNA editing patterns across four different tissues from TBG and IMG. To the best of our knowledge, this is the first comprehensive investigation of high-altitude adaptation-related RNA editing.

We systematically identified the RNA editing events in goats and obtained 84,132 high-quality RESs, their non-redundant sites (10,842) were distributed unevenly among the chromosomes, which is consistent with the study in bovine and yak [18,42]. In addition, we also found that A-to-I editing (62.61%) was the most common type of RNA editing, which has been confirmed in sheep, yaks, pigs, and bovines [11,18,19,42]. Interestingly, C-to-U editing (19.26%) sites were ranked second in goats but were not observed in chickens due to the absence of *APOBEC1* [36], which caused a decrease in the abundance of edited mRNA in the mouse intestine [43]. This indicates that C-to-U editing may play an important role in goats, and the type of editing mechanism varies among species. Moreover, those A-to-I and C-to-U editing sites were mainly enriched in the GO terms associated with cell energy metabolism, cytoskeleton, and cellular activity.

RNA editing events were mediated by their editing catalyzing genes and related to editing-related gene expression. Previous studies show that *ADAR1* and *APOBEC1* play vital roles in RNA editing and transcript stability [44], and they also control C2C12 myoblasts and muscle development [45,46,47]. In this research, we found that 32.5% of RESs were significantly correlated with their catalyzing gene expression, while the remaining RESs may be affected by regulators such as *AIMP2* [48], *Pin1,* and *WWP2* [49] or other regulatory networks [50]. In addition, 53.3% of A-to-I editing sites were correlated with the *ADAR* family, and C-to-U editing is primarily catalyzed by *APOBEC1* [51,52] because no evidence has supported the editing activity of *APOBEC2* until now. We could not draw a reliable conclusion about the correlation between the editing levels of RESs and the expression levels of RNA editing catalyzing genes among tissues because no significant trends were observed. This may be caused by the following factors: (1) We found no obvious tissue-specific manner of RNA editing catalyzing genes (except *APOBEC4*) in the heart, kidney, lung and longissimus dorsi; (2) RNA editing is affected by other regulators, but we found less tSESs; (3) the limited of tissue types. It is worth noting that in our research, C-to-U editing sites correlated with the expression of *APOBEC1* and *APOBEC2*, which amounted to 94 and 104, respectively, suggesting that *APOBEC2* may have editing activities. Since RNA editing may be relevant to editing-related genes [19], we found that 39.1% PCGs and 40.5% RESs were significantly correlated, and their correlation coefficient ranged from 0.3 to 0.8. These PCGs, such as *SFTPB* [53,54], *ATP2A1* [55], *EIF2AK4* [56], *KLF13* [57], *LDB3* [58], *SLC13A1* [59], and *ACMSD* [60], with organizational functions may have potential roles in environmental adaptation.

Additionally, the specific sequence around RNA editing sites might have an impact on the RNA structure and its editing patterns. In accordance with previous studies carried out in chickens [36], pigs [21], bovines [18,61], and yaks [62], one base upstream of the A-to-I editing site depleted G, whereas one base downstream enriched G, which demonstrated that ADAR prefers to target sequences of neighboring nucleotides [63,64]. We also observed that the probabilities of A-to-I and C-to-U editing for the 10 bp flanking on each side were different (C-to-U editing had more Gs and As, while A-to-I editing had more Gs and Cs). These findings suggest that the goat A-to-I editing and C-to-U editing patterns may be affected by different flanking nucleotides of the edited A and C, respectively. According to our data, the encoded aminos were different after A-to-I or C-to-U editing, and their physical and chemical properties were changed. Importantly, we observed that the minimal distances from RESs to exons were obviously close to 0, suggesting the RESs could contribute in alternative splicing of transcripts by disrupting splicing signals or creating new splice sites, thus expanding the genetic information of the genome. In addition, C-to-U editing had lower minimal distances to exons than A-to-I editing, indicating that C-to-U editing might have higher activity on alternative splicing than A-to-I editing. These results suggested that A-to-I editing and C-to-U editing may use different mechanisms in goat RNA editing.

Comparing RNA editing profiles across tissues (heart, lung, kidney, and longissimus dorsi muscle) from TBG and IMG, we realized there were little differences in the distribution of A-to-I and C-to-U editing sites between TBG and IMG, which come from a common ancestor. Consistent with a previous study, more A-to-I and C-to-U sites were edited in the lung (1669) and kidney (1526) tissues and fewer in the muscles (heart 1221 and longissimus dorsi 1419) [65]. In addition, A-to-I and C-to-U editing levels were significantly higher in the kidney of TBG than that of IMG. The A-to-I editing in the longissimus dorsi muscle between TBG and IMG also showed a significant difference, coinciding with the study that environmental stress alters RNA editing patterns [28]. Numerous studies have found that RNA editing varies across tissues and may be caused by the following factors: (1) the differential distribution of RNA editing deaminase in tissues [22]; (2) the differential expression levels of the transcripts being edited in tissues [65]; and (3) it can also be interpreted as the lung and kidney tissues being more effective than the muscles during altitude adaption [66,67,68]. These findings indicate that the RNA editing pattern in tissues has a relatively larger difference than in populations. It is worth noting that the TBG’s kidney has higher editing levels than IMG’s, revealing RNA editing in special tissues, which may be related to TBG’s adaptation to the plateau’s high cold and hypoxia.

RNA editing sites vary across tissues, such as A-to-I events, which are dominantly tissue-dependent [11,65]. Our findings indicate that 18, 272, 68, and 176 RESs appeared exclusively in the heart, kidney, lung, and longissimus dorsi muscles, respectively. Skeletal muscle owns the largest proportion (47.6%) of high correlation for the editing level of A-to-I tSESs and its edited gene expression compared to other organs. Moreover, a previous study reported that the RNA editing sites at nonsynonymous sites showed adaptation signals in honeybees [69]. In this research, 85.7% A-to-I and 71.4% C-to-U nonsynonymous sites cause amino acid changes in the kidney, 84.2% A-to-I and 91.7% C-to-U nonsynonymous sites in the longissimus dorsi muscle may potentially cause physiological and pathological changes in organisms [3,4,5]. Moreover, five muscle-specific tSESs genes (*MYH8*, *MYH2*, *MYH3*, *NEB*, and *OBSCN*) were significantly enriched in muscle functions. These results confirm that tissue-specific RNA editing patterns and functions exist in goats.

By characterizing the functional aspect of RESs, we anchored 41 and 29 population-specific editing sites (pSESs) in TBG and IMG, respectively, and 53 population-differential editing sites (pDESs). Their corresponding genes, such as *ZNF644* [70], *ATF6* [71,72], and *ADAM10* [73] in TBG, *MYCT1* [74,75], *OBSCN* [76], and *OGFOD1* [77] in IMG, as well as pDESs-edited genes *MTO1* [78,79], *ARAP2* [80], *SCP2* [81], and *KIF17* potentially regulate the body’s homeostasis to adapt to environmental stress [82]. We also observed that 19 sites that correlated with 16 edited genes were related to energy metabolisms such as ATP binding molecular function, translation, and immune response [83,84,85]. Furthermore, we found 73.3% population-differential, 73.2% TBG-specific, and 80% IMG-specific were nonsynonymous A-to-I sites that changed protein products, which showed strong adaptation signals. These findings suggest that the RNA-editing genes may contribute to mammalian adaptation to the environment.

## 4. Materials and Methods

### 4.1. Sample Collection

To attain an excellent experimental control for the high-altitude TBG living in Rutog county of the Qinghai-Tibet Plateau (4500 m altitude, 12% O_2_ content), we analyzed the origin, performances (including body weight, fleece yield, and fleece thickness), and genetic distances among goat populations. Finally, the IMG living in Alxa of Inner Mongolia grassland (1200 m altitude, 18% O_2_ content) were anchored since they performed closely and shared a common ancestor with TBG [39,40,41]. We randomly chose five healthy one-year-old female goats from each group (n = 5). Then they were humanely sacrificed, and their blood from the jugular vein and four tissues (heart, lung, kidney, and longissimus dorsi muscle) were quickly sampled and stored in liquid nitrogen accordingly.

### 4.2. DNA and RNA Sequencing

Genomic DNA was extracted from blood samples using a DNA extraction kit (Tiangen, Beijing, China). Then, a high-throughput sequencing library with an average length of 80 insert fragments of 350 bp was constructed for DNA sequencing by double-terminal 10× with a 150 bp reading length using an Illumina HiSeq platform sequencing instrument (Illumina, San Diego, CA, USA).

Total RNA was extracted from tissue samples using the Trizol method (Ambion, Austin, TX, USA) and then quantified by a NanoPhotometer spectrophotometer (IMPLEN, Munich, Germany). The mRNAs enriched by polyA magnetic beads from qualified RNA samples were used to construct the library. Finally, an Illumina HiSeq 2500 sequencing platform (Illumina, San Diego, CA, USA) was employed for double-terminal 300 bp (2 × 150 bp) high-pass sequencing, and clean reads amounted to 10 G per library.

### 4.3. Reads Alignment and Variant Calling

Trimming: Genomic DNA and RNA raw data were analyzed using Trimmomatic v0.39 to filter connectors and remove readings containing more than 10% unknown bases or reading segments containing more than 50% low-quality (Qphred ≤ 20) bases.

Alignment: Using BWA v0.7.17, we aligned DNA clean reads to the ARS1 goat reference genome (refgenome) retrieved from an Ensembl database (release-95). In addition, RNA clean reads were mapped to the same refgenome by HISAT2 v2.1.0 after generating an index on the goat refgenome ARS1 using hisat2-build. It was based on the known gene annotation file (GTF, release-95), extracting the splicing sites and exons coordinated from refgenome by Python scripts extract_splice_sites.py and extract_exons.py with default parameters. Then we used default-set hisat2 to align the sequence and refgenome.

BAM process: DNA and RNA alignment BAMs were sorted by SAMtools v1.8.0 with their genome coordinates. To avoid potential PCR or sequence optical artifacts, we removed duplicated reads mapped to the exact location by the MarkDuplicates tool in GATK v4.1.8.0. FixMateInformation function in GATK was further used to ensure that all matching sequencing read paragraphs were synchronized.

Variation calling: To reduce the random systematic error caused by sequencing instruments, we retrieved goat single nucleotide polymorphisms (SNPs) from published databases, including Ensemble (release-95) and variant call format (VCF). We then combined them using BaseCalbrator and ApplyBQSR in GATK v4.1.8.0 for each sequence alignment file by recalibrating the base quality score. Then, HaplotypeCaller in GATK was used to call variation, and SelectVariants was employed for separating SNP and short fragment insertion and deletion (INDEL). No filter process was performed in this step to obtain the complete genomic variation results, allowing the strictest control of subsequent RESs identified.

### 4.4. Identifying RESs

RESs Detection: Using REDItools v1.2.1 [86], we detected RESs based on a strategy of comparative genomic DNA and transcriptome RNA sequence. REDItoolDnaRna.py function in REDItools detects candidate RESs by detecting the base mismatch between DNA and RNA sequence point by point and counts base distribution, coverage depth, mean mass fraction, and frequency of variation.

RESs Filtration: The selectPositions.py in REDItools was firstly used to filter candidate RESs with default parameters, including -c 10 -C 5 -v 2 -V 0 -f 0.01 -F 0.95 -e -u, denoting that: (1) RNA sequence reads covered that site at least ten times; (2) DNA sequence reads covered that site at least five times; (3) Minimum support reads of editing base is 2; (4) The number of bases supporting DNA sequence mutations is 0; (5) Minimum mutation proportion of RNA sequence is 0.01; (6) Minimum non-mutant proportion of DNA sequence is 0.95; (7) Eliminates multiple base mutations at the same site. Moreover, the unqualified candidate RESs were excluded with the following conditions: (1) Overlapped with the known SNPs or INDELs retrieved from the Ensemble database (release-95) via intersectBed function of BEDtools v2.17.0; (2) Detected in less than three samples; (3) With editing level amounted to 1 in all detected samples; (4) Located at sex chromosomes or unplaced in refgenome. All downstream analyses were based on the obtained high-quality RESs.

Note: The RNA sequencing library in this study was a ploy(A) strategy and non-strand specific. Thus, we cannot precisely determine which strand occurred in RNA editing events. Therefore, both AG bases change that appeared in positive-stranded genes and TC in reverse-stranded genes reported by REDItools were regarded as A-to-I editing. Similarly, CT in positive-stranded genes and GA in reverse-stranded ones were assigned as C-to-U editing.

### 4.5. Gene Annotation and Enrichment Analysis

Based on the goat ARS1 genome annotation file (Ensembl, ARS1, GTF, release-95), the intersectBed function of BEDtools v2.27.1 was employed to intersect RESs to gene feature regions, including exons, introns, 5′-UTR, 3′-UTR, and intergenic regions. To investigate the functions of candidate RESs genes, we performed Gene Ontology (GO) enrichment and Kyoto Encyclopedia of Genes and Genomes (KEGG) pathway analyses using the Fisher test, based not on the poorly annotated goat genome but on the human database (org.Hs.eg.db v 3.14.0). Meanwhile, the functions of those Ensembl genes without official symbols were investigated through UniProt (www.uniprot.org, accessed on 12 March 2021), a high-quality database of protein sequences and functional information (UniProt 2021). Ensembl gene IDs were converted to Entry IDs to obtain their homologous proteins’ functions. STRING (www.string-db.org, accessed on 6 March 2021) was used for constructing protein-protein interaction networks based on human annotation.

All *p*-values were adjusted by Benjamini and Hochberg’s (BH) algorithm, and the significance level was set at 0.05.

### 4.6. Sequence Preference

R package ggseqlogo was used to plot the frequency of bases. A total of 21-bp sequences containing each RES and its flanking 10-bp upstream and downstream region in the goat genome were extracted. We selected editing sites in CDSs to obtain the codon sequence and counted the distance from the start codon after removing the non-coding region. We then divided the accepted coding length by 3, to attain the codon’s position. Codons coded the edited amino changed A to G or C to T at those positions. For those RESs located in the negative sense, we converted them to complementary antisense sequences.

### 4.7. Population and Tissue Comparisons of RESs

Tissue-specific editing sites (tSESs): To begin, we compared RESs in four organs collected from each goat individual. Then, ANOVA in R v4.0.2 was used to compare the difference in editing level between tissues, and the significance threshold was set to *p* < 0.05. With five samples collected per tissue, the RESs exclusively appeared in only one tissue and were defended as tSESs.

Population-differential editing sites (pDESs): We filtered out high-quality RESs supported by at least three samples in a tissue per population to compare the RNA editing profiling between populations unbiasedly. Then pDESs were defined as those RESs passing the Wilcox test, representing significantly different editing levels between groups in the same tissue. The *p*-values were adjusted via the padjust function in R v4.0.2 with Benjamini and Hochberg’s (BH) algorithm, and the threshold was set to 0.05.

Population-specific editing sites (pSESs): Those sites altered in a specific group with at least three samples supported per tissue were defined as pSESs. For example, if an RNA editing event appeared in the kidney of three IMG individuals but failed to be identified in the TBG samples, we defined it as an IMG pSESs in the kidney.

### 4.8. Calculation of Gene Expression

We quantified the gene expression using stringtie v1.3.4d for calculating read counts based on alignment BAM files and a genome coordinate file for each gene characteristic (Ensembl, ARS1, GTF, release-95). After generating the gene count matrix, we normalized data using CalcNormFactors and cpm functions in the R v4.0.2 package edgeR v3.38.2 and then calculated gene expression using RPKM (Reads Per Kilobase Million) method. Differential expression genes were detected using the exactTest process in edgeR v3.38.2, with BH adjusted *p*-values < 0.05 and log2 fold change (log2FC) > 1 as a significant threshold.

### 4.9. Correlation Analysis between Gene Expression and RNA Editing Level

The correlation between RNA editing level and the expression of its editing-related gene for each RESs was detected for 40 individuals using the R function correlation (cor) with the Spearman method and significance threshold set to *p*-value < 0.05.

## 5. Conclusions

We identified thousands of RESs in the heart, lung, kidney, and longissimus dorsi muscle from TBG and IMG and characterized their potential functions, such as recoding protein and alternative splicing. Furthermore, our findings revealed that the majority tSESs, pSESs, and pDESs showed adaptive signals in the tissues and populations. This will be helpful to animal breeders and biomedical researchers studying plateau-related diseases.

## Figures and Tables

**Figure 1 ijms-24-03115-f001:**
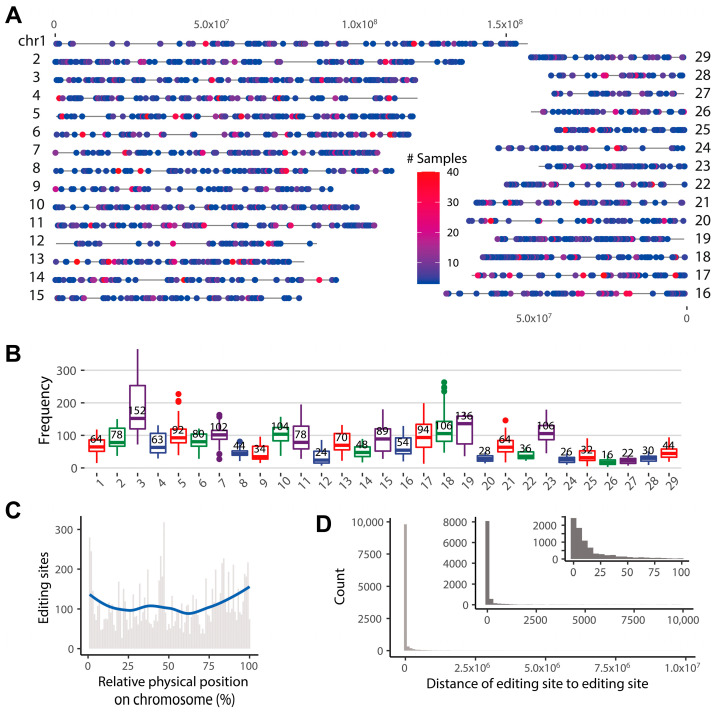
Characteristics of RNA editing sites (RESs) across goat genome. (**A**) Genome distribution of RESs. (**B**) Frequency distribution of RESs across the chromosomes. (**C**) Location distribution of RESs on chromosomes. (**D**) The count of RESs of the distance from editing site to editing site.

**Figure 2 ijms-24-03115-f002:**
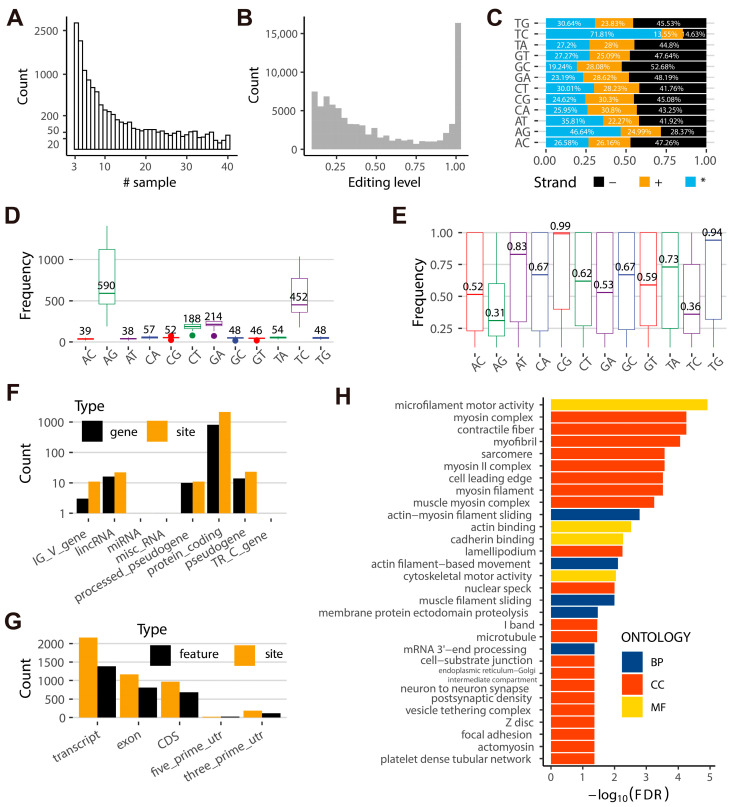
Gene annotation and enrichment analysis of A-to-I and C-to-U editing. (**A**) The count of RESs across goat individuals. (**B**) The editing levels of RESs. (**C**) The proportion of 12 RNA editing types in the DNA double-strand, +: DNA positive strand, −: DNA reverse strand, *: ambiguous strand. (**D**) The median count of RESs across RNA editing types. (**E**) The editing levels of RESs across RNA editing types. (**F**) Gene biotype of A-to-I and C-to-U editing sites and genes. (**G**) The position of A-to-I and C-to-U editing sites in different regions of a gene. (**H**) GO enrichment of A-to-I and C-to-U editing-related protein-coding genes. BP: Biological process, MF: Molecular function, CC: cellular component.

**Figure 3 ijms-24-03115-f003:**
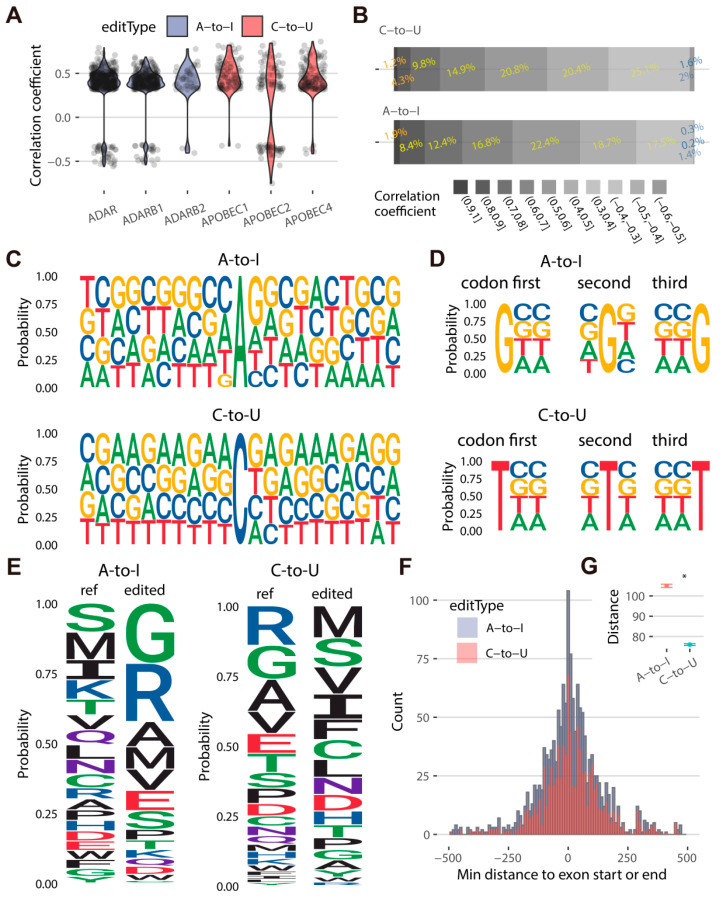
Functional characterization of goat A-to-I and C-to-U editing. (**A**) The correlation coefficient between the editing levels of A-to-I or C-to-U sites and the expression levels of catalyzing genes (*ADAR*, *ARARB1*, *ADARB2*, *APOBEC1*, *APOBEC2*, and *APOBEC4*). (**B**) The correlation coefficient between the editing levels of A-to-I or C-to-U sites and the expression of RNA editing-related genes. (**C**) Sequence preferences for base flanking (−10, +10) A-to-I and C-to-U editing sites detected. The letter height indicates the level of preference or depletion. (**D**) Codon sequence preferences of A-to-I and C-to-U editing sites. The letter height indicates the level of preference or depletion. (**E**) The encoded amino acids changed after A-to-I and C-to-U editing. ref indicates reference amino acid (unedited), edited indicates A-to-I and C-to-U edited amino acid. (**F**) The distribution of distance from A-to-I or C-to-U sites to exon start or end. (**G**) Comparing the minimal distances to the exon boundary of A-to-I and C-to-U sites to assess alternative splicing activity. * indicates *p* < 0.05.

**Figure 4 ijms-24-03115-f004:**
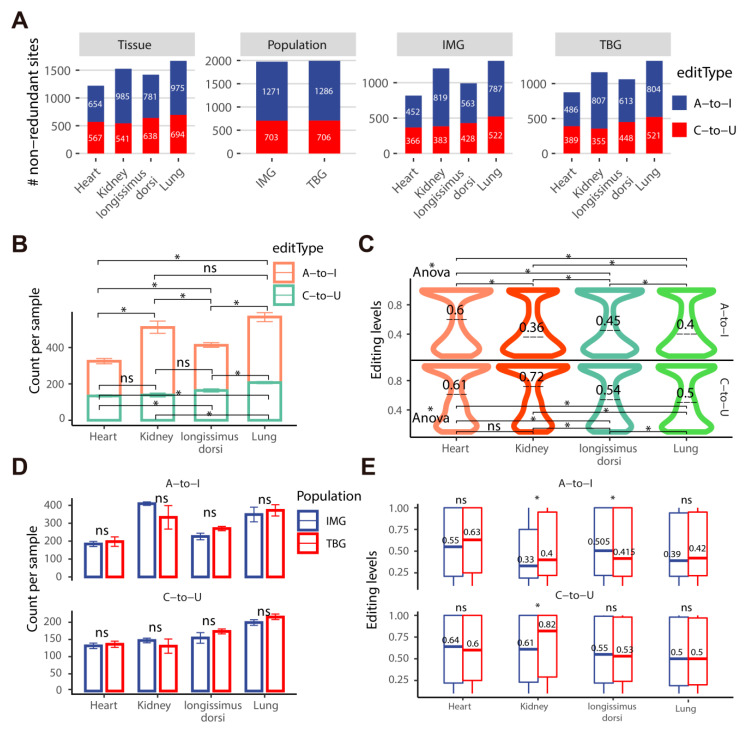
The numbers and editing levels of RESs across goat tissues and populations. (**A**) The numbers of A-to-I and C-to-U editing sites across tissues and populations. (**B**) The numbers of A-to-I and C-to-U editing sites per sample across tissues. (**C**) Distribution of A-to-I and C-to-U editing levels across tissues. (**D**) The numbers of A-to-I and C-to-U editing sites per sample between TBG and IMG populations. (**E**) The editing levels of A-to-I and C-to-U editing sites between TBG and IMG populations. TBG: Tibetan cashmere goats, IMG: Inner Mongolia cashmere goats, ns, no significant difference (*p* > 0.05), * *p* < 0.05.

**Figure 5 ijms-24-03115-f005:**
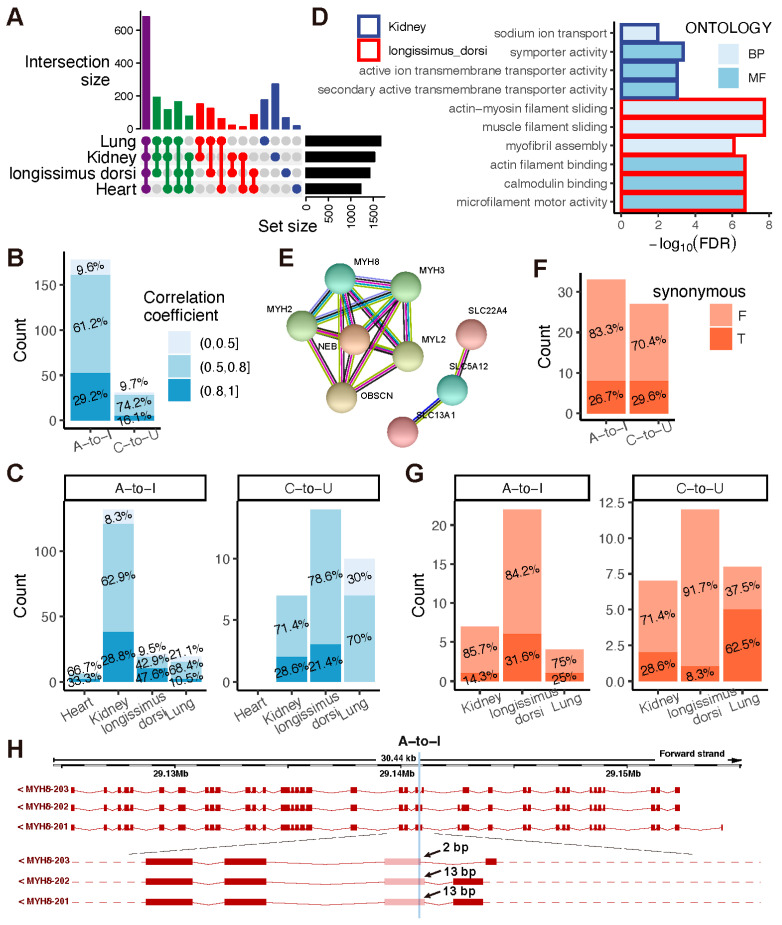
Tissue-specific RNA editing sites (tSESs) in goats. (**A**) The numbers of shared RESs across goat tissues. (**B**) Correlation between tSESs-related genes and A-to-I and C-to-U editing. (**C**) Correlation between tSESs-related genes and A-to-I and C-to-U editing across tissues. (**D**) GO enrichment of tSESs-related genes. (**E**) Three tSESs-related genes in the kidney and five in the longissimus dorsi muscle with specific tissue functions. (**F**) Protein coding changes after tissue-specific A-to-I and C-to-U editing. (**G**) Protein-coding changes after tissue-specific A-to-I and C-to-U editing among tissues. (**H**) The genome location of A-to-I tSESs on *MYH8*. BP: Biological process, MF: Molecular function, F: False, non-synonymous, T: True, synonymous.

**Figure 6 ijms-24-03115-f006:**
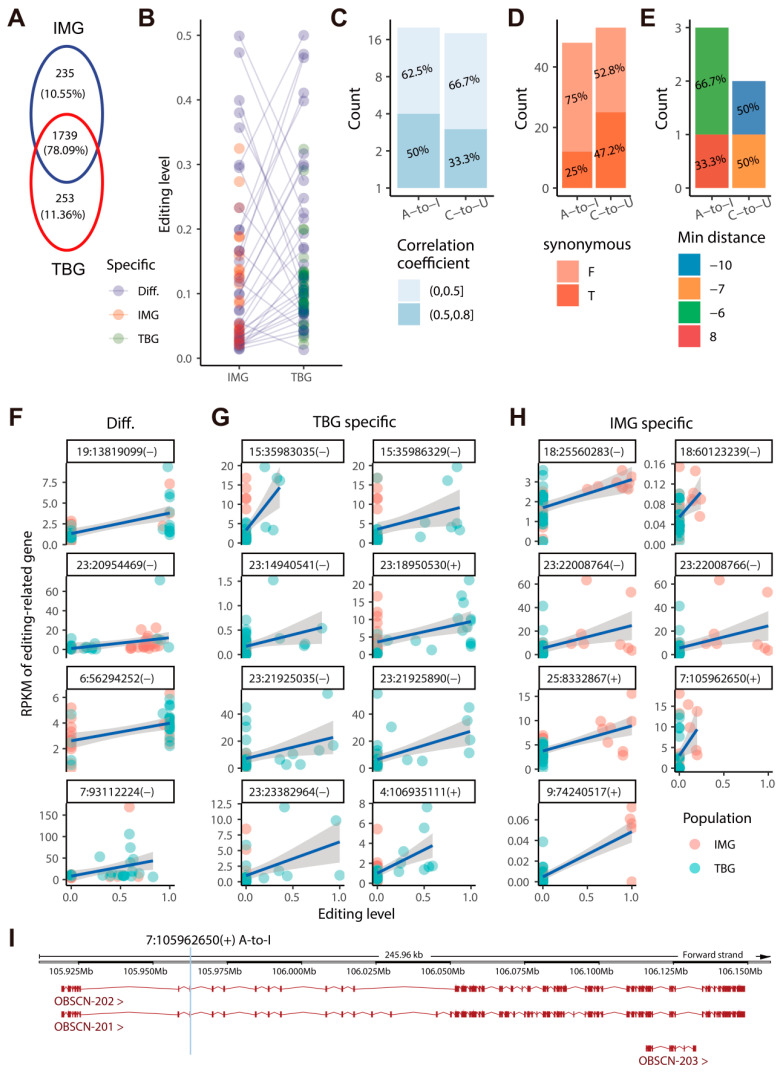
Population-specific RNA editing sites (pSESs) in goats. (**A**) Sharing numbers of RESs among IMG and TBG populations. (**B**) Population-differential editing sites (pDESs) among IMG and TBG. (**C**) Correlation between the pSESs and pDESs-related genes and A-to-I and C-to-U editing. (**D**) Protein-coding changes after population-specific and differential A-to-I and C-to-U editing. (**E**) Distribution of min distance to the exon of pSESs and pDESs. (**F**–**H**) Correlation between expression levels of editing-related genes and editing levels of pDESs (**F**), TBG pSESs (**G**), and IMG pSESs (**H**). (**I**) The annotated IMG pSESs editing-related gene OBSCN in the goat genome. F: False, non-synonymous, T: True, synonymous.

## Data Availability

The raw sequencing data are available through the NCBI data accession number PRJNA925004.

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
