# Peer review of "The Profiles and Functions of RNA Editing Sites Associated with High-Altitude Adaptation in Goats"

_ijms, 2023, doi:10.3390/ijms24043115_

Round 1
Reviewer 1 Report
Dear authors,
This work analyzes the RNA editing sites in heart, lung, kidney, and muscle from two goat breeds adapted to high-altitude. The results are very interesting, and reveal a lot of non-redundant editing sites, mainly in the kidney. The paper is well written and structured, the introduction provides sufficient background and include relevant references, the cited references are relevant to the research, the research design is appropriate, the methods are adequately described, the results are clearly presented, and the conclusion are supported by the results.
Only, minor changes are necessary:
- Several figures need to improve its quality. Concretely, figures 2C, and figure 3, to improve the understand.
- In figure 4, if the statistical differences are p<0,05, p-values < 0,01 do not have different significance.
- In figure 4C, the Anova seems to statistical test, but in the material and methods section, Anova test does not appear.
- Figure 6: change the font size.
Author Response
Dear Reviewer:
We greatly appreciate your comments and suggestions on our manuscript. We revised our manuscript accordingly to address your critiques. All changes are marked as yellow in the revised version, which we would like to submit for your kind consideration.
- Several figures need to improve its quality. Concretely, figures 2C, and figure 3, to improve the understand.
Re: For this suggestion, we have provided high-quality figures in the manuscript, and we also revised the legend of Figure 2C and Figure 3 to improve the understanding, as follows:
Figure 2C. The proportion of 12 RNA editing types in the DNA double strand, +: DNA positive strand, −: DNA reverse strand, *: ambiguous strand. (in p.4 line 116-118)
Figure 3. Functional characterization of goat A-to-I and C-to-U editing. A. The correlation coefficient between the editing levels of A-to-I or C-to-U sites and the expression levels of catalyzing genes (ADAR, ARARB1, ADARB2, APOBEC1, APOBEC2, and APOBEC4). B. The correlation coefficient between the editing levels of A-to-I or C-to-U sites and the expression of RNA editing-related genes. C. Sequence preferences for base flanking (−10, +10) A-to-I and C-to-U editing sites detected. The letter height indicates the level of preference or depletion. D. Codons sequence preferences of A-to-I and C-to-U editing sites. The letter height indicates the level of preference or depletion. E. The encoded amino acids changed after A-to-I and C-to-U editing. ref indicate reference amino acid (unedited), edited indicate A-to-I and C-to-U edited amino acid. F. The distribution of distance from A-to-I or C-to-U sites to exon start or end. G. Comparing the minimal distances to the exon boundary of A-to-I and C-to-U sites to assess alternative splicing activity. ns (no significant difference) indicates P > 0.05, * indicates P < 0.05. (in p.6 line 170-181)
- In figure 4, if the statistical differences are p<0,05, p-values < 0,01 do not have different significance.
Re: We have adjusted the significance threshold to P < 0.05, ns (no significant difference) indicates P > 0.05, * indicates P < 0.05. Moreover, we also have changed the significance difference to * in the figure 3G and figure 4. (in p.7 line 206)
- In figure 4C, the Anova seems to statistical test, but in the material and methods section, Anova test does not appear.
Re: We have added the information of Anova in the material and methods section, as follow:
Then, ANOVA in R v4.0.2 was used to compare the difference of editing level between tissues, and the significance threshold was set to P < 0.05. (in p.15 line 487-489)
- Figure 6: change thefont size.
Re: We have changed the 12 font to 9 font of Figure 6 legend. (in p.11 line 277-283)
Reviewer 2 Report
The Li Li et al manuscript "The profiles and functions of RNA editing sites associated with high-altitude adaptation in goats" is an high-quality of paper to describe the potential correlation between RNA modification and the environmental adaptation. It is a comprehensive paper with detailed bioinformatic analysis for the genome-wide RNA editing. I have only one minor comments for this manuscript. The expressions of RNA editing enzymes are tissue/organ specific, so is it possible to link the correlation between specific RNA editing enzyme with RNA editing expression level (or Condon sequence preferences) in tissue/organ-specific manner.
Author Response
Dear Reviewer:
We greatly appreciate your comments and suggestions on our manuscript. We revised our manuscript accordingly to address your critiques. All changes are marked as yellow in the revised version, which we would like to submit for your kind consideration.
- The expressions of RNA editing enzymes are tissue/organ specific, so is it possible to link the correlation between specific RNA editing enzyme with RNA editing expression level (or Condon sequence preferences) in tissue/organ-specific manner.
Re: Thank you for your suggestion. We have analyzed the correlation coefficient between the editing levels of RESs and the expression levels of RNA editing catalyzing genes among individuals, as shown in Figure 3A and 3B. We calculated the correlation between the expressions of RNA editing enzymes and RNA editing levels for each tissue according to your suggestion. However, no significant trends are observed, likely due to 1) We found no obvious tissue-specific manner of RNA editing enzymes (except APOBEC4) in the heart, kidney, lung and longissimus dorsi; 2) RNA editing is affected by other regulators, but we found less tissue-specific RNA editing sites in the heart (18), kidney (272), lung (68), and longissimus dorsi muscle (176); 3) the limited of tissue types. For example, in Figure 4C&4E and Figure S1, the A-to-I editing level was highest in the heart, whereas ADAR, ADARB1 and ADARB2 expression were lowest; while in lung, ADAR and ADARB1 expression levels were highest, with the editing level was not the lowest. Furthermore, the editing level of C-to-U was highest in the kidney, whereas the expression level of APOBEC2 and APOBEC4 was low. The editing level of C-to-U was lowest in the lung, although APOBEC4 was highest. Therefore, we could not draw a reliable conclusion on whether there is a tissue-specific manner between the expression of RNA editing enzymes and the levels of RNA editing.
We have added this part to the discussion section of the manuscript, as follows:
We could not draw a reliable conclusion about the correlation between the editing levels of RESs and the expression levels of RNA editing catalyzing genes among tissues, because no significant trends was observed. This may be caused by the following factors: 1) We found no obvious tissue-specific manner of RNA editing enzymes (except APOBEC4) in the heart, kidney, lung and longissimus dorsi; 2) RNA editing is affected by other regulators, but we found less tSESs; 3) the limited of tissue types. (in p.12 line 312-318)
Figure S1 was showed in the cover letter.
Figure S1 Gene expressions of ADAR, ADARB1, ADARB2, APOBEC1, APOBEC2, and APOBEC4 in tissues
